# Targeting the KRAS Oncogene for Patients with Metastatic Colorectal Cancer

**DOI:** 10.3390/cancers17091512

**Published:** 2025-04-30

**Authors:** Ruoyu Miao, James Yu, Richard D. Kim

**Affiliations:** 1Department of Hematology and Oncology, Winship Cancer Institute, Emory University, Atlanta, GA 30322, USA; ruoyu.miao@emory.edu; 2Department of Gastrointestinal Oncology, H. Lee Moffitt Cancer Center and Research Institute, Tampa, FL 33612, USA; james.yu@moffitt.org

**Keywords:** colorectal cancer, KRAS, mutation, targeted therapy, resistance

## Abstract

KRAS mutations occur in approximately 40% of colorectal cancer (CRC) cases. These mutations drive tumorigenesis through the constitutive activation of key signaling pathways, contributing to therapeutic resistance and poor prognosis. The development of KRAS G12C inhibitors has shown promise in clinical trials. However, their efficacy is limited to a small subset of KRAS-mutant CRC, and resistance mechanisms often emerge through compensatory pathway activation. This review provides a comprehensive overview of the molecular mechanisms, current advances and challenges, and future prospects in the management of KRAS-mutant CRC.

## 1. Introduction

Colorectal cancer (CRC) is one of the most common cancers worldwide and a leading cause of cancer-related mortality [1,2]. The Kirsten rat sarcoma viral oncogene homolog (KRAS) gene, a member of the RAS family, encodes a small GTPase protein that acts as a critical molecular switch in cellular signaling pathways regulating proliferation, differentiation, and survival [3,4]. Mutations in KRAS occur in approximately 40% of CRC cases, with the most frequent alterations found in codons 12, 13, and 61 [3,5,6]. These mutations result in the constitutive activation of KRAS, leading to unregulated downstream signaling through pathways such as RAS-RAF-MEK-ERK (MAPK) and PI3K-AKT-mTOR, which drive tumorigenesis [4,7]. Clinically, KRAS mutations are associated with poor prognosis, resistance to anti-EGFR monoclonal antibodies (like cetuximab and panitumumab), and limited therapeutic options, making this a challenging subset of CRC to treat [3,6,8].

Despite the historically “undruggable” nature of KRAS, recent advances in molecular biology and targeted drug development have sparked new hope for addressing this mutation. The approval of KRAS G12C inhibitors, such as sotorasib and adagrasib, has demonstrated that the direct targeting of mutant KRAS is feasible. Beyond small-molecule inhibitors, novel therapeutic strategies are emerging, including immunotherapy (e.g., immune checkpoint inhibitors (ICIs), KRAS neoantigen vaccines, and adoptive T-cell therapies), RNA interference (RNAi) to silence mutant KRAS expression, and CRISPR-based gene editing to correct or disrupt oncogenic KRAS alleles [9,10,11,12,13,14,15,16]. Additionally, nanoparticle delivery systems are being explored to enhance drug bioavailability and tumor-specific targeting while minimizing off-target effects [17,18]. Among these approaches, small-molecule inhibitors remain particularly advantageous because of their high efficacy rates in blocking KRAS signaling, oral bioavailability, and relatively lower production costs compared to those of biologics or cell-based therapies. Their well-established pharmacokinetic profiles and ease of combination with existing regimens further underscore their clinical utility [19,20].

This review aims to explore the molecular mechanisms underlying KRAS-mutant CRC, recent progress in understanding its biology, and emerging therapeutic strategies. By highlighting cutting-edge research and clinical advancements, this article seeks to provide a comprehensive update on this critical topic in oncology.

## 2. Molecular Biology of KRAS Mutations

The KRAS gene, located on chromosome 12p12.1, encodes a small GTPase that functions as a molecular switch in key cellular signaling pathways (Figure 1) [4,5]. Under normal physiological conditions, KRAS cycles between an active, GTP-bound state and an inactive, GDP-bound state. This cycling is tightly regulated by guanine nucleotide exchange factors (GEFs), such as SOS1, which promote GTP binding, and GTPase-activating proteins (GAPs), such as NF1, which accelerate GTP hydrolysis [7,21,22]. In its active state, KRAS interacts with downstream effectors, including RAF kinases, phosphoinositide 3-kinase (PI3K), and Ral guanine nucleotide exchange factors (RalGEFs), to regulate cellular processes, such as proliferation, differentiation, and survival [4,22,23].

In CRC, mutations in KRAS are among the earliest genetic events [24] and impair the intrinsic GTPase activity of KRAS, rendering it as constitutively active. This leads to the persistent activation of downstream signaling pathways, even in the absence of upstream growth factor stimuli [25,26]. The RAS-RAF-MEK-ERK (MAPK) pathway is a primary effector of KRAS signaling, promoting uncontrolled cellular proliferation and survival [27,28]. Simultaneously, the PI3K-AKT-mTOR pathway is activated, contributing to metabolic reprogramming, resistance to apoptosis, and enhanced tumor growth [29,30]. The oncogenic potential of KRAS mutations is further amplified by crosstalk with other signaling pathways, including the Wnt/β-catenin and transforming growth factor-β (TGF-β) pathways [31,32,33]. KRAS mutations also induce alterations in cellular metabolism, favoring glycolysis and glutaminolysis to meet the energetic and biosynthetic demands of rapidly proliferating tumor cells [34,35,36]. Furthermore, mutant KRAS contributes to genomic instability by promoting reactive oxygen species (ROS) production and impairing DNA repair mechanisms [37,38]. These molecular and cellular alterations not only drive tumor initiation and progression but also create a tumor microenvironment (TME) conducive to immune evasion and resistance to therapy [36,37]. Thus, understanding the molecular biology of KRAS is essential for developing effective therapeutic strategies.

KRAS mutations most commonly occur in codons 12, 13, and 61, with G12D, G12V, and G13D being the most frequently observed in CRC [39,40]. The functional impact of KRAS mutations in CRC depends on the specific amino acid substitution, as different mutations result in distinct biochemical and signaling alterations that influence tumor behaviors and therapeutic responses [40]. For example, G12D mutations predominantly activate the PI3K pathway, contributing to enhanced survival signaling, while G12V mutations are more dependent on the RAF-MEK-ERK cascade, promoting aggressive tumor growth [40,41]. Unlike other KRAS mutations, which are generally resistant to anti-EGFR therapies, G13D mutations retain some sensitivity because of impaired binding to NF1 [42,43]. These functional differences influence prognoses, metastatic potentials, and responses to targeted therapies. Understanding the unique properties of KRAS mutations is crucial for developing tailored therapeutic strategies.

## 3. Biological Implications of KRAS Mutations

### 3.1. Tumor Initiation and Progression

KRAS mutations play a pivotal role in the multistep progression of colorectal cancer. Following the inactivation of APC, mutant KRAS drives the transition from the normal colonic epithelium to premalignant adenomas by promoting uncontrolled cellular proliferation and resistance to apoptosis. The deletion of chromosome 18q and inactivation of TP53 further accelerate tumor progression and promote the transition to malignancy [44,45,46,47]. The oncogenic effects of KRAS are not limited to cell-autonomous mechanisms but also involve interactions with the TME, facilitating angiogenesis and immune evasion [36].

### 3.2. Impact on the TME

KRAS mutations significantly remodel the TME, creating conditions that support tumor growth and metastasis. KRAS-driven tumors secrete vascular endothelial growth factor (VEGF), promoting angiogenesis and enhancing nutrient and oxygen delivery to the tumor [48,49]. Mutant KRAS upregulates the production of cytokines and chemokines, such as IL-6, CXCL8, and TGF-β, which recruit regulatory T-cells (Tregs), myeloid-derived suppressor cells (MDSCs), and tumor-associated macrophages (TAMs) to the TME. These immune cells suppress antitumor immunity and enable immune evasion [7,26,48,50]. Additionally, KRAS mutations upregulate programmed death-ligand 1 (PD-L1) expression on tumor cells, inhibiting T-cell-mediated cytotoxicity [26]. These changes collectively foster an immunosuppressive TME that is permissive to tumor progression and resistant to therapy.

### 3.3. Role in Metastasis

KRAS mutations are strongly associated with the metastatic potential of CRC. Mutant KRAS promotes epithelial–mesenchymal transition (EMT), a process characterized by the loss of epithelial cellular adhesion and increased cellular motility, and suppresses immune surveillance to allow circulating tumor cells to colonize distant sites. This enables cancer cells to invade surrounding tissues and disseminate to distant organs [51,52,53].

### 3.4. Therapeutic Resistance

One of the most clinically significant implications of KRAS mutations in CRC is their association with therapeutic resistance. KRAS mutations confer primary resistance to anti-EGFR monoclonal antibodies, such as cetuximab and panitumumab, which are effective only in RAS wild-type tumors. This resistance arises because mutant KRAS drives downstream signaling independent of upstream EGFR activation, rendering EGFR inhibition as ineffective [30,54,55]. KRAS mutations also contribute to resistance to chemotherapy and radiotherapy. For example, mutant KRAS alters the apoptotic machinery, making tumor cells less susceptible to DNA damage induced by cytotoxic agents [30]. Similarly, KRAS-driven metabolic reprogramming enhances the survival of cancer cells under therapeutic stress [35,56,57].

In summary, the biological implications of KRAS mutations in CRC are multifaceted, encompassing tumor initiation, progression, metastasis, immune evasion, and treatment resistance. These insights into the molecular and cellular effects of KRAS mutations provide a foundation for developing innovative therapeutic approaches to improve outcomes for patients with KRAS-mutant CRC.

## 4. Advances in Therapeutics

### 4.1. KRAS G12C Inhibitors

KRAS G12C mutations occur in approximately 3–4% of CRCs [58,59]. Recent advancements in targeted therapy have led to the development of small-molecule, covalent KRAS G12C inhibitors, such as sotorasib and adagrasib, which selectively and irreversibly bind to the mutant KRAS G12C protein, locking it in an inactive GDP-bound state and turning off its oncogenic signaling.

The phase 1 CodeBreaK 100 trial evaluated oral sotorasib (AMG 510) in 129 patients with advanced solid tumors harboring the KRAS G12C mutation. A total of 56.6% had treatment-related adverse events (TRAEs) of any grade, and 11.6% had grade 3 or 4 TRAEs. A total of forty-two patients had CRC and had previously received at least two lines of systemic therapy. The study observed an objective response rate (ORR) of 7.1% (three patients with a confirmed partial response (PR)) and a disease control rate (DCR) of 73.8% (thirty-one patients) in the CRC subgroup. The median progression-free survival (PFS) was 4.0 months (range: 0.0–11.1+) among the CRC patients [60]. In the single-arm, phase 2 CRC cohort of the CodeBreaK 100 trial, 62 patients with KRAS G12C-mutant advanced CRC, whose disease had progressed after fluoropyrimidine, oxaliplatin, and irinotecan, were enrolled and received sotorasib monotherapy (960 mg once daily). Six patients achieved a partial response, with an ORR of 9.7% (95% confidence interval (CI): 3.6–19.9) and median duration of response (DOR) of 4.2 months (interquartile range (IQR): 2.9–8.5). The DCR was 82.5% (95% CI: 70.5–90.8). The median PFS and overall survival (OS) were 4.0 months (95% CI: 2.8–4.2) and 10.6 months (95% CI: 7.7–15.6), respectively. TRAEs of any grade occurred in 55% of the patients, most commonly diarrhea (21%) and nausea (16%). A total of 10% developed grade 3 TRAEs, including diarrhea (3%), and 2% had grade 4 TRAEs, with elevated blood creatinine phosphokinase [61].

Adagrasib (MRTX849) was investigated in a phase 1/2 KRYSTAL-1 trial [62,63]. A total of 44 heavily pretreated patients with metastatic CRC (mCRC) harboring mutant KRAS G12C received adagrasib monotherapy (600 mg orally, twice daily). The ORR was 19% (95% CI: 8–33), with a median DOR of 4.3 months (95% CI: 2.3–8.3). The median PFS and OS were 5.6 months (95% CI: 4.1–8.3) and 19.8 months (95% CI: 12.5–23.0), respectively. TRAEs of any grade occurred in 93% of the patients, most commonly diarrhea (66%), nausea (57%), vomiting (45%), and fatigue (45%). A total of 34% of the patients developed grade 3 or 4 TRAEs, including anemia (9%) and diarrhea (7%) [63].

Divarasib (GDC-6036) is a second-generation, covalent KRAS G12C inhibitor that selectively and irreversibly locks the protein in its inactive state. It has been shown to be 5–20 times as potent and up to 50 times as selective compared to sotorasib and adagrasib, as per in vitro studies [64,65]. In a phase 1 study, divarasib (50–400 mg orally, once daily) was evaluated in 137 patients with advanced or metastatic solid tumors harboring the KRAS G12C mutation. TRAEs occurred in 93% of the patients, most commonly nausea (74%), diarrhea (61%), and vomiting (58%). Grade 3 events occurred in 11% of the patients and included diarrhea (4%), an increase in the alanine aminotransferase (ALT) level (3%), and an increase in the aspartate aminotransferase (AST) level (3%). A grade 4 TRAE (an anaphylactic reaction) occurred in one patient (1%). Among the 55 patients with CRC, a confirmed response was observed in 29.1% of the patients (95% CI: 17.6–42.9), with a median DOR of 7.1 months (95% CI: 5.5–7.8). The median PFS was 5.6 months (95% CI: 4.1–8.2). Of the 39 patients who received divarasib at 400 mg, the confirmed ORR was 35.9% (95% CI: 21.2–52.8), with a median DOR of 7.7 months (95% CI: 5.7 to could not be estimated). The median PFS was 6.9 months (95% CI: 5.3–9.1) [64]. The study also demonstrated that the rapid and deep decline in the circulating tumor DNA (ctDNA) fraction was associated with the treatment response and PFS [66].

Olomorasib (LY3537982) is another potent and highly selective second-generation KRAS G12C inhibitor. It achieves a high target occupancy rate at a very low-dose exposure [67,68]. The phase 1/2 LOXO-RAS-20001 study (NCT04956640) is evaluating olomorasib in patients with KRAS-G12C-mutant advanced solid tumors. The updated results showed that, among the 157 patients who received olomorasib monotherapy (50–200 mg orally, twice daily), TRAEs of any grade occurred in 62% of the patients, most commonly diarrhea (24%), fatigue (10%), and nausea (10%). A total of 5% had grade ≥ 3 TRAEs. Of the 32 patients with CRC, the ORR was 9% (3 PR), and the DCR was 84%. The mPFS was 4 months (95% CI: 3–7). Olomorasib also demonstrated efficacy in patients with non-small cell lung cancer (NSCLC) who had prior exposure to a KRAS G12C inhibitor [67].

### 4.2. KRAS G12C Inhibitors Combined with Anti-EGFR Therapy

Although KRAS G12C inhibitors have shown promise in early-phase clinical trials, studies like KRYSTAL-1 and CodeBreaK 100 have highlighted modest response rates, emphasizing the need for combination strategies. Although KRAS G12C inhibitors suppress the mutant KRAS protein, the receptor tyrosine kinase (RTK, primarily EGFR in CRC)-mediated upstream feedback reactivation of the RAS-MAPK-signaling pathway may occur in a KRAS-G12C-independent manner, resulting in treatment resistance [69,70]. Preclinical data have suggested that the combinatorial targeting of EGFR and KRAS G12C leverages a dual-pathway blockade to inhibit tumor growth more effectively and may overcome the adaptive resistance to KRAS G12C inhibition [70].

CodeBreaK 101 (NCT04185883) is an ongoing phase 1 trial exploring sotorasib monotherapy and in combination with other anti-cancer therapies, including anti-EGFR agents (panitumumab), MEK inhibitors, and ICIs, in advanced solid tumors harboring the KRAS G12C mutation. The combination of sotorasib (960 mg once daily) and panitumumab was evaluated in patients with chemotherapy–refractory KRAS-G12C-mutant mCRC. A total of 40 patients were enrolled in the dose-expansion cohort. The confirmed ORR was 30.0% (95% CI: 16.6–46.5). The median PFS was 5.7 months (95% CI: 4.2–7.7). The median OS was 15.2 months (95% CI: 12.5–not estimable (NE)) [71]. The phase 3 CodeBreaK 300 trial randomized 160 patients with KRAS-G12C-mutant, chemotherapy–refractory mCRC to receive sotorasib (960 mg) plus panitumumab, sotorasib (240 mg) plus panitumumab, or the investigator’s choice of trifluridine–tipiracil or regorafenib (the standard of care (SOC)). The primary endpoint was PFS, and the trial was not powered to detect a difference in OS because of the low prevalence of the KRAS G12C mutation in metastatic CRC. Grade ≥ 3 TRAEs occurred in 35.8%, 30.2%, and 43.1% of the patients, respectively. Skin-related toxicity and hypomagnesemia were the most common adverse events observed with sotorasib–panitumumab. The median PFSs were 5.6 months (95% CI: 4.2–6.3; hazard ratio (HR) 0.49; *p* = 0.006, vs. SOC), 3.9 months (95% CI: 3.7–5.8; HR 0.58; *p* = 0.03, vs. SOC), and 2.2 months (95% CI: 1.9–3.9), respectively [72]. With a median follow-up of 13.6 months, the median OS was not reached (HR 0.70; *p* = 0.20, vs. SOC) vs. 11.9 months (95% CI: 7.5-NE; HR 0.83; *p* = 0.50, vs. SOC) vs. 10.3 months (95% CI: 7.0-NE). The ORRs were 30.2% (95% CI: 18.3–44.3), 7.5% (95% CI: 2.1–18.2), and 1.9% (95% CI: 0.0–9.9), respectively [73]. On January 16, 2025, the U.S. Food and Drug Administration (FDA) approved sotorasib (recommended dosage: 960 mg) with panitumumab for adult patients with KRAS-G12C-mutant mCRC, who had received prior fluoropyrimidine-, oxaliplatin-, and irinotecan-based chemotherapies. The phase 3 CodeBreaK 301 trial is comparing sotorasib + panitumumab + FOLFIRI versus FOLFIRI with or without bevacizumab in the first-line setting [74].

The combination of adagrasib (600 mg twice daily) and cetuximab was also evaluated in the phase1/2 KRYSTAL-1 trial, with the primary endpoint of the ORR. A total of 94 patients received the combination therapy. A total of 16% developed grade 3–4 TRAEs. Among the 28 evaluable patients, the ORR was 34.0%, the DCR was 85.1%, and the median DOR was 5.8 months (95% CI: 4.2–7.6). The median PFS and OS were 6.9 months (95% CI: 5.7–7.4) and 15.9 months (95% CI: 11.8–18.8), respectively. All the patients developed TRAEs, with grade 3–4 in 27.7% of the patients. Exploratory analyses suggested that ctDNA may be associated with responses and acquired resistance [75]. On 21 June 2024, the FDA granted accelerated approval to adagrasib plus cetuximab for adults with KRAS-G12C-mutant locally advanced or mCRC after prior fluoropyrimidine-, oxaliplatin-, and irinotecan-based chemotherapies. This combination is being compared with SOC chemotherapy as second-line treatment in the ongoing phase 3 KRYSTAL-10 trial [63].

Divarasib in combination with cetuximab was evaluated in patients with metastatic KRAS-G12C-positive CRC (*n* = 29) in a phase 1 study. All (100.0% of) the patients experienced at least one TRAE, most commonly rash (96.6%), diarrhea (82.8%), nausea (72.4%), and vomiting (48.3%). Grade 3 TRAEs occurred in 37.9% of the patients and grade 4 in 6.9%. The confirmed ORR was 62.5% (95% CI: 40.6–81.2) in KRAS-G12C-inhibitor-naïve patients (*n* = 24), with a median DOR of 6.9 months (95% CI: 5.6-NE). The median PFS was 8.1 months (95% CI: 5.5–12.3) [65].

In the combination cohort of the phase 1 study (NCT04956640), 46 patients with treatment-refractory mCRC were treated with olomorasib (100 or 150 mg twice daily) and cetuximab. Dermatitis acneiform (59%), diarrhea (44%), dry skin (44%), hypomagnesemia (33%), and fatigue (30%) were among the most common treatment-emergent adverse events (TEAEs). Among the 38 patients evaluable for efficacy, the ORR was 42% (all the PR), and the DCR was 95% [68].

The encouraging efficacy with a manageable toxicity profile supports the further investigation of divarasib and olomorasib combined with cetuximab in KRAS-G12C-positive CRC. Ongoing clinical trials that investigate KRAS G12C inhibitors and combination therapies in KRAS-G12C-mutant CRC are summarized in Table 1.

A similar strategy has been explored in the management of mCRC harboring the BRAF V600E mutation, which occurs in 8–12% of CRC cases and confers an aggressive clinical course. Single-agent BRAF inhibitors (e.g., vemurafenib) have shown limited efficacy in mCRC because of EGFR-mediated feedback activation [76]. The BEACON CRC trial demonstrated that the combination of encorafenib and cetuximab (with or without binimetinib) provided a survival benefit over that of standard therapy in previously treated BRAF-V600E-mutant mCRC [77]. More recently, the phase 3 BREAKWATER study has been evaluating encorafenib + cetuximab ± chemotherapy (FOLFOX or FOLFIRI) in first-line and later-line settings. The early results suggest promising efficacy, with a notable increase in the ORR to 60.9% in the triplet therapy arm versus 40.0% with chemotherapy alone (odds ratio 2.443; *p* = 0.0008) in chemotherapy-naïve patients, reinforcing the potential of BRAF/EGFR inhibition as a backbone in earlier treatment lines [78]. The findings led to the FDA’s accelerated approval of this combination on 20 December 2024, establishing a new standard of care in first-line treatment for BRAF-V600E-mutant mCRC.

### 4.3. Targeting Other KRAS Mutations

#### 4.3.1. KRAS G12D

Unlike KRAS G12C, KRAS G12D lacks a reactive residue proximal to the switch-II-binding pocket, making a covalent modification challenging [79,80]. The development of KRAS-G12D-specific inhibitors utilized different approaches. MRTX1133 is a potent, highly selective, and non-covalent KRAS G12D inhibitor that binds to the GDP-bound, inactive form of KRAS G12D and occupies the switch-II pocket. It markedly inhibited KRAS-dependent signaling and demonstrated potent antitumor activity both in vitro and in vivo [79,81]. A phase 1/2 trial of MRTX1133 in advanced solid tumors harboring KRAS G12D is underway.

HRS-4642 is a high-affinity, selective, non-covalent KRAS G12D inhibitor. It exhibited high selectivity in inhibiting the growth of KRAS-G12D-mutant cell lines and demonstrated robust efficacy against KRAS-G12D-mutant human pancreatic cancer and CRC models [80]. The preliminary results of the first in-human, phase 1 study in patients with advanced KRAS-G12D-mutant solid tumors reported a tolerable safety profile. Among the 13 efficacy-evaluable patients, 61.1% had stable disease, and 33.3% experienced target lesion shrinkage, including those with lung cancer and CRC [82].

LY3962673 is another non-covalent KRAS G12D inhibitor with high affinity for KRAS G12D-GDP and demonstrated robust antitumor activity in cancer cell lines and multiple KRAS-G12D-mutant patient-derived xenograft (PDX) models [83]. RMC-9805 is a first-in-class, mutant selective covalent inhibitor of RASG12D(ON), which is the active, GTP-bound state (“ON” state) of RASG12D. It disrupts downstream RAS signaling by the steric occlusion of effector binding and inhibits cellular proliferation and apoptosis. Combinations of RMC-9805 with either RMC-6236 (RASMULTI(ON) inhibitor) or an anti-EGFR antibody improved the depth of the response and delayed the onset of resistance in KRAS G12D CRC models [84]. These and several other KRAS-G12D-targeted inhibitors are being evaluated in early-phase trials, which enrolled patients with mCRC, as monotherapy and in combination therapy (Table 2).

Proteolysis-targeting chimeras (PROTACs) are bifunctional molecules that recruit E3 ubiquitin ligases, e.g., VHL or CRBN, to tag oncogenic proteins, like KRAS, for ubiquitination and subsequent proteasomal degradation [85,86]. Unlike direct KRAS inhibitors, PROTACs achieve complete protein knockdown, potentially overcoming resistance mechanisms driven by compensatory feedback loops [87]. Early preclinical studies have demonstrated that KRAS-targeting PROTACs, including degraders specific for KRAS G12D and KRAS G12V mutants, effectively reduce KRAS protein levels and suppress downstream MAPK signaling in various cancer cell lines and xenograft models [88,89]. The preliminary results of a phase 1 trial evaluating a first-in-class, KRAS-G12D-selective protein degrader, ASP3082, in advanced pancreatic cancer, CRC, and NSCLC showed an acceptable safety profile and promising antitumor activity, especially in pretreated pancreatic cancer [90].

#### 4.3.2. Pan-KRAS Inhibitors

Pan-KRAS inhibitors are a promising area of research for treating KRAS-mutant mCRC. QTX3034 is a highly selective, non-covalent, multi-KRAS inhibitor with potent activity across several KRAS variants, including G12D and G12V [91,92]. QTX3034 binds to GDP-bound forms of mutant and wild-type KRAS, inhibiting KRAS signaling in vitro and inducing tumor regressions in both pancreatic and colorectal KRAS G12D xenograft models. A phase 1 trial in KRAS-G12D-mutant tumors is ongoing [92]. RMC-6236 is a non-covalent, potent tri-complex inhibitor of mutant and wild-type RAS variants and targets the active, GTP-bound state of RAS(ON) [93]. RMC-6236 exhibited robust anticancer efficacy across RAS-addicted cell lines and multiple tumor types in KRAS G12X xenograft models [94]. These and several other pan-RAS inhibitors are being investigated in KRAS-mutant mCRC (Table 2).

### 4.4. Targeting KRAS-Signaling Pathways

KRAS mutations activate downstream signaling cascades, including the MAPK (RAS-RAF-MEK-ERK) and PI3K-AKT-mTOR pathways, which drive tumor growth and survival. Novel therapies targeting these pathways aim to overcome the limitations of monotherapy by disrupting resistance mechanisms or amplifying therapeutic effects.

SOS1 is a GEF that facilitates the exchange of GDP for GTP and KRAS activation. MRTX0902 is a selective and potent SOS1 inhibitor that disrupts the KRAS:SOS1 protein–protein interaction. MRTX0902 synergized with adagrasib and augmented the antitumor activity in KRAS-G12C-mutant human NSCLC and CRC xenograft models. In KRAS-MAPK-pathway-mutant models, the dual inhibition of RTK/MAPK pathway signaling by MRTX0902 with EGFR or RAF/MEK inhibitors resulted in greater suppression of pathway signaling and better antitumor responses [95]. MRTX0902 alone and with adagrasib are being evaluated in a phase 1/2 study (Table 1).

SHP2 is a protein tyrosine phosphatase essential for activating the RAS-MAPK pathway downstream of receptor tyrosine kinases. The inhibition of SHP2 disrupts KRAS signaling and sensitizes tumors to KRAS inhibitors [96,97]. Trials evaluating SHP2 inhibitor monotherapy and in combination with an KRAS inhibitor are ongoing (Table 1, Table 2 and Table 3). The preliminary result of glecirasib (a KRAS G12C inhibitor) plus JAB-3312 (an SHP2 inhibitor) demonstrated a manageable safety profile in patients with KRAS-G12C-mutant tumors, including 17 CRCs, and a promising ORR and a promising PFS as a front-line treatment for patients with KRAS G12C NSCLC [98].

MEK inhibitors, such as trametinib and binimetinib, block the downstream MAPK-signaling cascade. MEK inhibitors in combination with various targeted therapies so far have resulted in limited clinical efficacy for KRAS-mutant mCRC, such as SAR405838 (an HDM2 inhibitor) plus pimasertib (an MEK1/2 inhibitor) [99], afatinib (a pan-HER inhibitor) plus selumetinib (an MEK inhibitor) [100], neratinib (a pan-ERBB inhibitor) plus trametinib [101], navitoclax (a Bcl-2 inhibitor) plus trametinib [102], panitumumab plus trametinib [103], and binimetinib plus palbociclib (a CDK4/6 inhibitor) [104]. Preclinical models suggest that combining MEK inhibitors with KRAS G12C inhibitors can reduce the feedback reactivation of the pathway and enhance tumor regression [69]. A phase 1/2 study is underway evaluating the KRAS G12C inhibitor JDQ443 in combination with trametinib for patients with advanced solid tumors, including mCRC (Table 1).

Similarly, although RAF inhibitors alone have shown limited success in KRAS-mutant CRC, combination strategies targeting RAF and MEK or RAF and EGFR are being explored to suppress MAPK signaling and potentially overcome resistance (Table 3). Avutometinib (VS-6766) is a unique RAF/MEK clamp that blocks MEK kinase activity and prevents MEK phosphorylation by RAF. The preclinical evaluation of avutometinib in combination with panitumumab showed significant antitumor activity in CRC PDX models harboring KRAS mutations, with more tumor regression observed in a KRAS G12V PDX model than in a KRAS G12D model [84].

The PI3K-AKT-mTOR pathway is often upregulated in KRAS-mutant tumors, driving resistance to KRAS inhibitors [105]. Targeting this pathway with PI3K inhibitors or dual PI3K/mTOR inhibitors is under investigation [106,107]. These agents have been tested in combination with KRAS inhibitors to simultaneously block multiple survival pathways in NSCLC [108,109].

### 4.5. Targeting Other Pathways

Synthetic lethality occurs when the inhibition of a secondary pathway in a tumor harboring a specific mutation (e.g., KRAS) results in cell death [110]. Synthetic lethality in KRAS-mutant mCRC represents a promising therapeutic strategy by targeting vulnerabilities specific to KRAS-mutant tumor cells while sparing normal cells. Because the direct inhibition of mutant KRAS has historically been challenging, researchers have focused on exploiting synthetic lethal interactions, such as targeting DNA damage repair pathways, key metabolic dependencies, and parallel signaling networks, in addition to inhibiting the KRAS effector pathways [111]. Understanding these interactions offers new avenues for overcoming drug resistance and improving outcomes in KRAS-mutant mCRC (Table 3).

For example, creatine kinase-B (CKB) has been identified as a cancer driver in KRAS-mutant CRC by promoting tumor growth and survival in hypoxia. The energetic metabolite phospho-creatine (PCr), generated by CKB, is imported into cells through the creatine transporter, SLC6A8. The small-molecule SLC6A8 inhibitor RGX-202-01 depletes intracellular PCr and ATP and leads to apoptosis. In a phase 1a study, RGX-202-01 monotherapy demonstrated objective antitumor activity for relapsed/refractory KRAS-mutant CRC without dose-limiting toxicity. RGX-202-01 in combination with FOLFIRI and bevacizumab was well tolerated in patients with pretreated CRC. Among nine patients with KRAS-mutant CRC, the ORR was 56%, and the DCR was 100% [112].

The serine/threonine kinase polo-like kinase 1 (PLK1) regulates the cell cycle by controlling mitotic entry and progression. The selective PLK1 inhibitor onvansertib has shown potent antitumor activity in CRC preclinical models, both as a single agent and in combination with irinotecan. Onvansertib inhibited the hypoxia pathway and demonstrated potent antitumor activity in combination with bevacizumab by suppressing angiogenesis. Additionally, the synthetic lethality between PLK1 inhibition and KRAS mutations has been observed in CRC models [113]. Onvansertib combined with FOLIFRI and bevacizumab exhibited manageable safety and promising efficacy as a second-line treatment in patients with KRAS-mutant mCRC [113,114].

### 4.6. Mutant-KRAS-Targeted Cancer Vaccines

Mutant-KRAS-targeted cancer vaccines are emerging as a promising immunotherapeutic strategy for KRAS-mutant mCRC. These vaccines aim to stimulate the immune system to recognize and attack cancer cells harboring specific neoantigens that are associated with KRAS mutations, such as G12D, G12V, or G12C. Peptide-based, mRNA, and dendritic cell vaccines are being explored to enhance T-cell responses [115,116]. Recent clinical trials have demonstrated the feasibility and safety of these approaches, with some showing early signs of efficacy [9,10,11]. For example, the cancer vaccine ELI-002 2P consists of the amphiphile (Amph) modification of G12D- and G12R-mutant KRAS peptides (Amph-Peptide-2P) and a CpG oligonucleotide adjuvant (Amph-CpG-7909), which enhances lymph node delivery and the immune response. In a phase 1 study, ELI-002 2P induced significant T-cell and biomarker responses in patients with immunotherapy-recalcitrant KRAS-mutant pancreatic and colorectal cancers with minimal residual disease (MRD) [10]. ELI-002 7P comprises amph-peptide 7P (G12X and G12D peptides) and amph-CpG-7909. In patients with pancreatic or colorectal cancer with MRD, ELI-002 7P induced higher median T-cell responses than ELI-002 2P at the recommended phase 2 dose and exhibited early indications of antitumor activity [11]. Challenges such as tumor immune evasion, antigen heterogeneity, and the immunosuppressive tumor microenvironment remain to be addressed to improve the vaccines’ efficacy [115]. Combination strategies with ICI or adoptive T-cell therapies may further enhance their therapeutic potential in mCRC (Table 4).

### 4.7. Adoptive T-Cell Therapy

Adoptive T-cell therapy (ACT) is another emerging approach that harnesses the immune system to specifically target and eliminate tumor cells expressing mutant KRAS. Strategies such as T-cell receptor (TCR)-engineered T-cells and tumor-infiltrating lymphocytes (TILs) are being explored to enhance the immune recognition of KRAS mutations (Table 4). It was reported that the adoptive transfer of mutant KRAS-G12D-specific, HLA-C*08:02-restricted TILs (CD8+ T-cells) achieved durable tumor regressions in one patient with mCRC [12]. Recent advancements in neoantigen discovery and gene-editing technologies, including CRISPR/Cas9, have facilitated the development of highly specific TCR-engineered T-cells that recognize mutant KRAS peptides presented by major histocompatibility complex (MHC) molecules [117,118].

## 5. Challenges and Future Perspectives

Despite the recent advancements in targeting KRAS-mutant CRC, several challenges remain in translating these breakthroughs into durable clinical benefits. One major hurdle is the heterogeneity of KRAS mutations, which differ in their biological activities and responses to therapies [40]. Additionally, it has been demonstrated that resistance can emerge following treatment with KRAS inhibitors, presenting a major hurdle to sustained therapeutic benefit. This resistance is driven by diverse mechanisms, including secondary mutations in KRAS that impair inhibitor binding; the activation of bypass-signaling pathways, such as PI3K-AKT and MAPK; and lineage plasticity, as exemplified by EMT. The feedback activation of RTKs, as well as the presence of co-mutations in TP53, STK11, or KEAP1, further complicate the therapeutic landscape [119,120]. Recent studies have also emphasized the roles of TME factors and immune modulation in shaping resistance [121]. Consequently, there is growing interest in combinatorial strategies that integrate KRAS inhibitors with agents targeting compensatory pathways, immune checkpoints, and epigenetic regulators to delay or overcome resistance [122].

Emerging evidence suggests that transcriptomic adaptations without acquired resistance mutations can contribute to resistance against anti-EGFR therapy in CRC [123]. Notably, KRAS mutations have been shown to induce substantial alterations in epigenetic modifications, which may also play a critical role in CRC progression [36]. Recent biomarker analysis from the KRYSTAL-1 trial identified acquired pathogenic alterations in 74% (25/34) of the cases [75], suggesting that DNA-driven resistance may represent a predominant mechanism of resistance to KRAS inhibitors, albeit with a small sample size. Although these findings underscore the genetic basis of resistance, the roles of epigenetic modifications in both primary and secondary resistances remain incompletely understood. Further investigations with larger cohorts are warranted to elucidate the interplay between epigenetic reprogramming and therapeutic resistance in KRAS-mutant CRC.

The exploration of synthetic lethality approaches targeting co-dependencies in KRAS-driven tumors, such as vulnerabilities in the DNA damage response or metabolic pathways, presents a novel strategy for therapeutic intervention [36,110,111]. Altered metabolism is a hallmark of KRAS-driven CRC, playing crucial roles in both tumor progression and resistance to therapy. Oncogenic KRAS mutations reprogram cellular metabolism to boost nutrient uptake, glycolysis, and glutamine utilization, thereby supporting rapid proliferation and survival under the nutrient-deprived and hypoxic conditions of the tumor microenvironment. These metabolic alterations create exploitable vulnerabilities, as targeting key pathways—such as glycolysis, the redox balance, and lipid metabolism—can selectively disrupt the growth of KRAS-mutant CRC cells while sparing normal tissues [35,124]. Emerging evidence has shown promise in the therapeutic potential of combining metabolic inhibitors with standard treatments to overcome drug resistance and enhance clinical outcomes [125]. A deeper understanding of KRAS-driven metabolic rewiring may pave the way for more effective, precision-targeted strategies in this aggressive CRC subtype.

Another major obstacle is the immunosuppressive TME commonly linked to KRAS mutations. KRAS-mutant CRC often exhibits low tumor mutational burden and poor immune cell infiltration, making it less responsive to ICIs [53]. This highlights the potential benefit of combining KRAS inhibitors with immunotherapy. However, recent clinical trials have highlighted significant toxicity concerns with this approach. Data from the CodeBreaK 100/101 and KRYSTAL-7 trials, which evaluated KRAS G12C inhibitors in combination with anti-PD-(L)1 antibodies, demonstrated a marked increase in grade 3–4 toxicities [126,127]. One potential hypothesis is that KRAS-targeted therapies modulate the immune system, thereby exacerbating immune-mediated toxicities driven by checkpoint inhibitors, though the precise mechanisms remain unclear [128]. To address these challenges, optimized combination strategies, including refining treatment sequencing, the intermittent dosing of KRAS inhibitors, and alternative combination regimens, can be explored to improve tolerability while maintaining efficacy [128].

Other immune-mediated KRAS-targeted approaches, including KRAS vaccines, T-cell therapy, and T-cell engagers, such as BiTEs-KRAS, are currently under investigation [128,129,130]. Additionally, targeting specific immune components, such as macrophages and regulatory T-cells, and immunosuppressive co-occurring mutations, like LKB1 and KEAP1 mutations, can be an area for further exploration [128,131,132]. These strategies could mitigate the immunosuppressive nature of KRAS-mutant cancers, including those of colorectal origin. Further investigations are warranted to refine these approaches and assess their clinical efficacy.

Looking ahead, a key focus will be the identification of biomarkers that can predict patient responses to KRAS-targeted therapies and support personalized treatment strategies. For example, ctDNA and liquid biopsies are emerging as valuable tools for tracking resistance mechanisms and adjusting treatments in real time by offering dynamic insights into tumor evolution and therapeutic responses [133,134,135]. Furthermore, multiomic approaches—integrating genomic, transcriptomic, and proteomic data—along with single-cell technologies, are anticipated to enhance our understanding of KRAS-driven biology and intratumoral heterogeneity, potentially uncovering novel therapeutic targets and effective drug combinations [136]. Innovative treatment modalities are also on the horizon, including RNAi to suppress mutant KRAS expression, CRISPR-based genome editing to modify or eliminate oncogenic KRAS alleles, nanoparticle-based delivery platforms designed to improve drug bioavailability and tumor specificity while minimizing off-target effects, and bifunctional PROTACs that harness E3 ubiquitin ligases to target KRAS for degradation via the proteasome [13,14,15,16,17,18,85].

## 6. Conclusions

KRAS mutations represent a critical driver of CRC, contributing to tumorigenesis, therapeutic resistance, and poor prognosis. Although historically deemed as undruggable, recent developments in KRAS-targeted therapies, including G12C inhibitors and combination approaches, have begun to shift the treatment paradigm for KRAS-mutant CRC. Despite these advances, the modest response rates, emergence of resistance, and limited treatment options for non-G12C mutations underscore the urgent need for continued innovation. Although significant hurdles remain, the growing understanding of KRAS biology and the rapid pace of translational research provide hope for meaningful progress in the management of KRAS-mutant CRC. A multidisciplinary approach integrating basic science, clinical innovation, and personalized medicine will be essential to overcome current challenges and improve survival and quality of life for these patients.

## Figures and Tables

**Figure 1 cancers-17-01512-f001:**
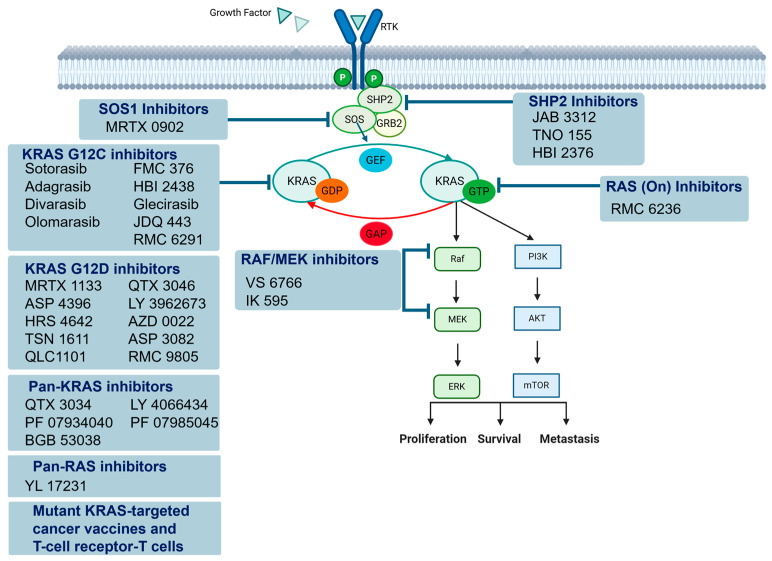
Schematic overview of the KRAS-signaling pathway and therapeutic targeting strategies in metastatic colorectal cancer. Contemporary approaches to KRAS-targeted therapy encompass both direct and indirect strategies aimed at suppressing RAS activation and downstream effector pathways, notably the MAPK and PI3K cascades. Direct inhibition includes mutation-specific agents, such as KRAS G12C and KRAS G12D inhibitors, as well as RAS(ON) inhibitors and other novel compounds currently under development. Indirect strategies involve targeting upstream modulators of KRAS activation, including SHP2 and SOS1 inhibitors, which interfere with the guanine nucleotide exchange process necessary for the transition from the inactive GDP-bound KRAS state to the active GTP-bound conformation. In parallel, the pharmacological inhibition of downstream effectors, particularly RAF and MEK, is also under active investigation.

**Table 1 cancers-17-01512-t001:** Selected clinical trials with KRAS G12C inhibitors in KRAS-G12C-mutant colorectal cancer.

Drug	Target	Clinical Trial	Phase	Intervention
Sotorasib (AMG 510)	KRAS G12C	NCT05198934 (CodeBreak 300)	3	Sotorasib 960 mg + panitumumab vs. sotorasib 240 mg + panitumumab vs. trifluridine/tipiracil or regorafenib
Sotorasib	KRAS G12C	NCT06252649 (CodeBreaK 301)	3	Sotorasib + panitumumab + FOLFIRI vs. FOLFIRI +/− bevacizumab
Adagrasib (MRTX849)	KRAS G12C	NCT03785249 (KRYSTAL-1)	1/2	Adagrasib +/− cetuximab
Adagrasib	KRAS G12C	NCT04793958 (KRYSTAL-10)	3	Adagrasib + cetuximab vs. mFOLFOX or FOLFIRI
Adagrasib	KRAS G12C	NCT05722327	1	Adagrasib + cetuximab + irinotecan
Adagrasib	KRAS G12C	NCT06412198	1/2	Adagrasib + cetuximab + cemiplimab
Adagrasib + TNO155	KRAS G12C, SHP2	NCT04330664 (KRYSTAL 2)	1	Adagrasib + TNO155
INCB099280 + adagrasib	PD-L1, KRAS G12C	NCT06039384	1	INCB099280 + adagrasib
KO-2806 +/- adagrasib	farnesyl transferase, KRAS G12C	NCT06026410	1	KO-2806 +/− adagrasib
MRTX0902 +/- adagrasib	SOS1, KRAS G12C	NCT05578092	1/2	MRTX0902 +/− adagrasib
Divarasib (GDC-6036)	KRAS G12C	NCT04449874	1	Divarasib +/− cetuximab
Olomorasib (LY3537982)	KRAS G12C	NCT04956640	1/2	Olomorasib +/− cetuximab or pembrolizumab
FMC-376	KRAS G12C	NCT06244771 (PROSPER)	1/2	FMC-376
HBI-2438	KRAS G12C	NCT05485974	1	HBI-2438
Glecirasib (JAB-21822)	KRAS G12C	NCT05194995	1/2	JAB-21822 + cetuximab
Glecirasib (JAB-21822) + JAB-3312	KRAS G12C, SHP2	NCT05288205	1/2	JAB-21822 + JAB-3312
JDQ443	KRAS G12C	NCT05358249	1/2	JDQ443 + trametinib, ribociclib, or cetuximab
JDQ443	KRAS G12C	NCT04699188 (KontRASt-01)	1/2	JDQ443 +/− TNO155 and/or tislelizumab
RMC-6291	KRAS G12C	NCT05462717	1	RMC-6291

**Table 2 cancers-17-01512-t002:** Selected clinical trials targeting other KRAS mutations in KRAS-mutant colorectal cancer.

Drug	Target	Clinical Trial	Phase	Intervention
MRTX1133	KRAS G12D	NCT05737706	1/2	MRTX1133
ASP4396	KRAS G12D	NCT06364696	1	ASP4396
HRS-4642	KRAS G12D	NCT06385678	1/2	HRS-4642 + adebrelimab, cetuximab, or SHR-9839 (EGFR/c-Met bispecific antibody)
TSN1611	KRAS G12D	NCT06385925	1/2	TSN1611
QLC1101	KRAS G12D	NCT06403735	1	QLC1101
QTX3046	KRAS G12D	NCT06428500	1	QTX3046 +/− cetuximab
LY3962673	KRAS G12D	NCT06586515 (MOONRAY-01)	1	LY3962673 +/− cetuximab or chemotherapy
AZD0022	KRAS G12D	NCT06599502	1/2	AZD0022 +/− cetuximab
ASP3082	KRASG12D degrader	NCT05382559	1	ASP3082 +/− cetuximab
RMC-9805	KRAS G12D(ON) inhibitor	NCT06040541	1	RMC-9805 +/− RMC-6236
QTX3034	Multi-KRAS inhibitor	NCT06227377	1	QTX3034 +/− cetuximab
PF-07934040	Pan-KRAS	NCT06447662	1	PF-07934040 +/− cetuximab or FOLFOX/bevacizumab
BGB-53038	Pan-KRAS	NCT06585488	1	BGB-53038 +/− tislelizumab or cetuximab
LY4066434	Pan-KRAS	NCT06607185	1	LY4066434 +/− cetuximab, chemotherapy, or pembrolizumab
PF-07985045, PF-07284892	Pan-KRAS, SHP2	NCT06704724	1	PF-07985045 +/− cetuximab, FOLFOX/bevacizumab, or PF-07284892
YL-17231	Pan-RAS	NCT06078800	1	YL-17231
Daraxonrasib (RMC-6236)	RAS-MULTI(ON) inhibitor	NCT05379985	1	RMC-6236
Daraxonrasib (RMC-6236)	RAS-MULTI(ON) inhibitor	NCT06445062	1	RMC-6236 +/− chemotherapy, cetuximab, bevacizumab and/or RMC-9805

**Table 3 cancers-17-01512-t003:** Selected clinical trials with other (non-KRAS) targeted therapies in KRAS-mutant colorectal cancer.

Clinical Trial	Phase	Intervention	Target
NCT05163028	1	HBI-2376	SHP2
NCT04121286	1	JAB-3312	SHP2
NCT05786924	1	BDTX-4933	RAF
NCT06194877	1	Brimarafenib (BGB-3245) + panitumumab	RAF, EGFR
NCT05200442	1/2	Avutometinib (VS-6766) + cetuximab	RAF/MEK, EGFR
NCT06270082	1	IK-595	RAF/MEK
NCT06634875	2	Isunakinra +/− pembrolizumab	IL-1R1
NCT06229340	2	leflunomide +/− MEK inhibitor and hydroxychloroquine +/- bevacizumab	DHODH, MEK
NCT03597581	1	Ompenaclid (RGX-202-01) +/− FOLFIRI or FOLFIRI/bevacizumab or FOLFOX/bevacizumab	SLC6A8
NCT03829410	1/2	Onvansertib + FOLFIRI + bevacizumab	PLK1
NCT04599140 (STOPTRAFFIC-1)	1/2	SX-682 +/− nivolumab	CXCR1/2

**Table 4 cancers-17-01512-t004:** Selected clinical trials with cancer vaccines and T-cell receptor (TCR) T-cells in KRAS-mutant colorectal cancer.

Clinical Trial	Phase	Intervention	Mechanism
NCT04117087	1	KRAS peptide vaccine + nivolumab + ipilimumab	Pooled mutant KRAS long peptide vaccine
NCT04853017 (AMPLIFY-201)	1	ELI-002 2P	KRAS G12D and G12R peptide vaccine
NCT05726864 (AMPLIFY-7P)	1/2	ELI-002 7P	KRAS/NRAS (G12D, G12R, G12V, G12A, G12C, G12S, and G13D) peptide vaccines
NCT06411691	1	SPL mKRASvax + balstilimab + botensilimab	Mutant KRAS long peptide vaccine
NCT06105021	1/2	AFNT-211	Autologous KRAS G12V-specific transgenic TCR T-cells
NCT06218914	1	NT-112	Autologous KRAS G12D-specific TCR T-cells
NCT06253520	1	KRAS TCR-transduced PBL + GRT-C903/GRT-R904	Autologous KRAS G12D- or KRAS G12V-specific TCR T-cells and KRAS vaccine
NCT06487377	1	IX001	Autologous KRAS G12D- or KRAS G12V-specific TCR T-cells
NCT06690281	2	KRAS TCR-transduced PBL	Autologous KRAS G12D- or KRAS G12V-specific TCR T-cells
NCT06707896	1	TCR1020-CD8 T-cells	Autologous KRAS G12V-specific TCR T-cells
NCT06767046	1	CRTKVA11	Autologous KRAS G12V-specific TCR T-cells

Abbreviations: PBL, peripheral blood lymphocytes. TCR, T-cell receptor.

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
