# Peer review of "Targeting the KRAS Oncogene for Patients with Metastatic Colorectal Cancer"

_cancers, 2025, doi:10.3390/cancers17091512_

Round 1
Reviewer 1 Report
Comments and Suggestions for Authors
I found the review very informative and well written.
I would just recommend to dedicate a paragraph to the comparision with BRAF mutations and patient management.
I would like the author to emphasize the research on kras PROTAC (including clinical trial)
Author Response
Thank you very much for taking the time to review this manuscript. We greatly appreciate the comments and suggestions from the reviewer. Please find the detailed responses below and the corresponding revisions highlighted in red in the re-submitted files.
Comment 1: I would just recommend to dedicate a paragraph to the comparison with BRAF mutations and patient management.
Response 1: We have added a paragraph discussing BRAF V600E-targeted treatment in mCRC in section 4.2. KRAS G12C Inhibitors Combined with Anti-EGFR Therapy (page 7, line 285). More detailed discussion in BRAF mutation-related management would be beyond the scope of this review.
Comment 2: I would like the author to emphasize the research on kras PROTAC (including clinical trial)
Response 2: We have added a paragraph discussing KRAS PROTAC in section 4.3.1. KRAS G12D (page 8, line 327).
Reviewer 2 Report
Comments and Suggestions for Authors
This review is well-written and addresses an important and timely topic. However, it would benefit from a clearer explanation of its significance compared to existing reviews on similar subjects. Highlighting what makes this review distinct—such as a novel angle, updated data, or unique integration of emerging strategies—would help position its contribution more strongly in the literature.
1.Please add a section on the future perspectives of KRAS-mutant colorectal cancer. This should include the current limitations in its treatment and potential solutions or strategies to overcome these challenges.
2. The introduction is too brief. It is recommended to expand it by including a brief overview of emerging therapeutic strategies such as immunotherapy, RNA interference, CRISPR-based gene editing, and nanoparticle delivery systems. Additionally, this would be a good place to highlight the advantages of small-molecule strategies, particularly their efficacy and cost-effectiveness.
3. The manuscript currently lacks any figures. It is suggested to add a figure summarizing the content of Section 3. A visual representation will enhance the clarity of the biological implications and make the overall concepts more straightforward and accessible to readers.
Author Response
Thank you very much for taking the time to review this manuscript. We greatly appreciate the comments and suggestions from the reviewer. Please find the detailed responses below and the corresponding revisions highlighted in red in the re-submitted files.
Comment 1: Please add a section on the future perspectives of KRAS-mutant colorectal cancer. This should include the current limitations in its treatment and potential solutions or strategies to overcome these challenges.
Response 1: We have accordingly revised section 5. Challenges and Future Perspectives (page 12, line 460, 464; page 13, line 486, 522).
Comment 2: The introduction is too brief. It is recommended to expand it by including a brief overview of emerging therapeutic strategies such as immunotherapy, RNA interference, CRISPR-based gene editing, and nanoparticle delivery systems. Additionally, this would be a good place to highlight the advantages of small-molecule strategies, particularly their efficacy and cost-effectiveness.
Response 2: Thank you for pointing this out. We have accordingly revised the introduction (page 2, line 46).
Comment 3: The manuscript currently lacks any figures. It is suggested to add a figure summarizing the content of Section 3. A visual representation will enhance the clarity of the biological implications and make the overall concepts more straightforward and accessible to readers.
Response 3: Thank you for this suggestion. We have created Figure 1 to show the KRAS signaling pathway, associated biological function, and therapeutic targeting strategies (page 2, line 74).
Reviewer 3 Report
Comments and Suggestions for Authors
This is a wonderful review article by Dr. Kim and colleagues that provides an in-depth discussion on the impact of targeting KRAS in colorectal cancer. The manuscript effectively highlights therapeutic opportunities and carries significant translational relevance. While the review is well-written and informative, a few key points need to be addressed before it is suitable for acceptance. These are outlined below:
1. It has been demonstrated that resistance can emerge following treatment with KRAS G12C and G12D inhibitors, driven by various underlying mechanisms (PMID: 37729426; PMID: 38975874). The authors should include a brief discussion on this important aspect.
2. Altered metabolism is known to play a significant role in KRAS-driven colorectal cancer, potentially offering a novel therapeutic window (PMID: 34227245, PMID: 36323004). The authors should include a brief discussion on this emerging area.
3. The authors should consider including a schematic model illustrating the therapeutic windows in KRAS-driven colorectal cancer, highlighting the various emerging therapeutic opportunities.
Author Response
Thank you very much for taking the time to review this manuscript. We greatly appreciate the comments and suggestions from the reviewer. Please find the detailed responses below and the corresponding revisions highlighted in red in the re-submitted files.
Comment 1: It has been demonstrated that resistance can emerge following treatment with KRAS G12C and G12D inhibitors, driven by various underlying mechanisms (PMID: 37729426; PMID: 38975874). The authors should include a brief discussion on this important aspect.
Response 1: Thank you for pointing this out. We have accordingly added this discussion in Challenges and Future Perspectives (page 12, line 464).
Comment 2: Altered metabolism is known to play a significant role in KRAS-driven colorectal cancer, potentially offering a novel therapeutic window (PMID: 33870211; PMID: 36323004). The authors should include a brief discussion on this emerging area.
Response 2: Thank you for pointing this out. We have accordingly added this discussion in Challenges and Future Perspectives (page 13, line 486).
Comment 3: The authors should consider including a schematic model illustrating the therapeutic windows in KRAS-driven colorectal cancer, highlighting the various emerging therapeutic opportunities.
Response 3: Thank you for this suggestion. We have created Figure 1 to show the KRAS signaling pathway, associated biological function, and therapeutic targeting strategies (page 2, line 74).
Round 2
Reviewer 3 Report
Comments and Suggestions for Authors
All concerns addressed, ready for acceptance.